# Latent Space Disentanglement in Diffusion Transformers Enables Precise Zero-shot Semantic Editing

## Abstract

Diffusion Transformers (DiTs) have recently achieved remarkable success in text-guided image generation. In image editing, DiTs project text and image inputs to a joint latent space, from which they decode and synthesize new images. However, it remains largely unexplored how multimodal information collectively forms this joint space and how they guide the semantics of the synthesized images. In this paper, we investigate the latent space of DiT models and uncover two key properties: First, DiT's latent space is inherently semantically disentangled, where different semantic attributes can be controlled by specific editing directions. Second, consistent semantic editing requires utilizing the entire joint latent space, as neither encoded image nor text alone contains enough semantic information. We show that these editing directions can be obtained directly from text prompts, enabling precise semantic control without additional training or mask annotations. Based on these insights, we propose a simple yet effective Encode-Identify-Manipulate (EIM) framework for zero-shot fine-grained image editing. Specifically, we first encode both the given source image and the text prompt that describes the image, to obtain the joint latent embedding. Then, using our proposed Hessian Score Distillation Sampling (HSDS) method, we identify editing directions that control specific target attributes while preserving other image features. These directions are guided by text prompts and used to manipulate the latent embeddings. Moreover, we propose a new metric to quantify the disentanglement degree of the latent space of diffusion models. Extensive experiment results on our new curated benchmark dataset and analysis demonstrate DiT's disentanglement properties and effectiveness of the EIM framework. [1]

## 1 Introduction

Diffusion models have achieved remarkable success in text-guided generation tasks, producing diverse, high-fidelity images and videos based on text prompts (Ma et al., 2024; Esser et al., 2024). These diffusion-based models are typically built on the UNet-based latent diffusion framework (Rombach et al., 2022), where input images are projected into latent embeddings and integrated with text embeddings through cross-attention layers within the UNet, as shown in Fig. 1(a.1). Recently, Diffusion Transformers (DiT) (Fig. 1(a.2)) introduced a new architecture that combines input image and text embeddings into a joint latent space and processes them through stacked self-attention layers. While empirical results suggest that this new structure enhances T2I controllability (Esser et al., 2024), it is unclear how multimodal information collectively forms this joint latent space and how these representations guide the semantics of the synthesized images.

A key to achieving controllability is the property of *semantic disentanglement*, where semantic attributes are encoded into distinct subspaces that can be independently controlled along specific editing directions. In other words, we can precisely manipulate a semantic attribute without affecting the others. This property has not been clearly observed in the latent space constructed in the UNet-based diffusion models (Lu et al., 2024; Kwon et al., 2022). Consequently, existing approaches alternatively rely on either loss-driven methods

---

[1] Our annotated benchmark dataset is publicly available at https://anonymous.com/anonymous/EIM-Benchmark.

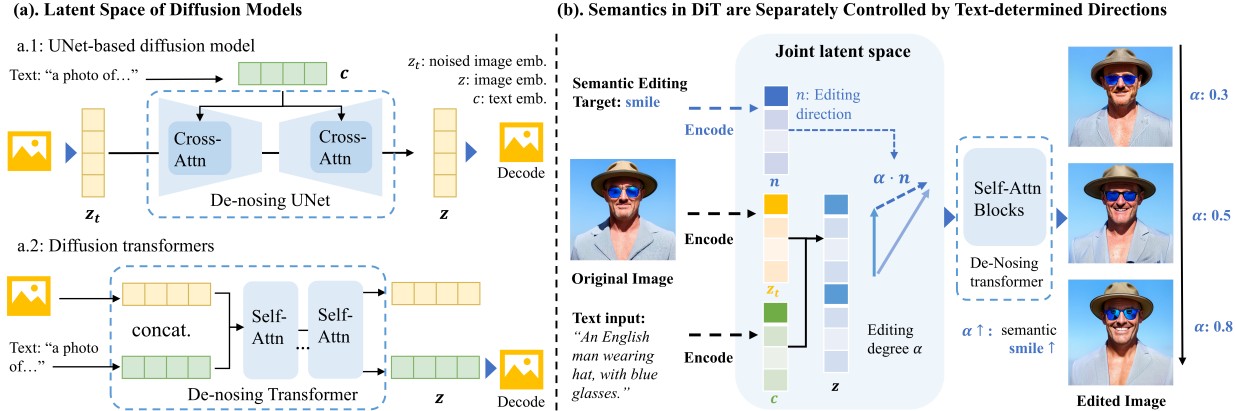

Figure 1: (a) UNet-based models align text embeddings with image embeddings via cross-attention layers. In contrast, DiT creates a joint latent space by combining the text embedding and image embedding, then feeds them into the denoising block. (b) DiT's has semantic disentangled latent space, where intensities of image semantics in generated images are controlled by separate directions, which can be easily identified.

to identify semantic control directions (Kim et al., 2022; Kwon et al., 2022; Avrahami et al., 2022; Mokady et al., 2023; Kawar et al., 2023) or attention maps to connect image semantics with text inputs (Hertz et al., 2022; Brooks et al., 2023). Note that these methods all require additional annotations (e.g., sub-image masks) or extensive optimization to achieve precise semantic control in image editing task, limiting their interpretability and generalizability in broader applications. In contrast, DiT's self-attention architecture projects image and text inputs into a joint latent space, potentially enabling direct link between image semantics and text prompts. Given this novel attention mechanism and DiT's remarkable text-to-image generation performance (Esser et al., 2024), we are motivated to delve into the fundamental research question: *Is the joint latent space of DiT semantically disentangled?*

By modifying image semantics in the joint latent space using directions from encoded text prompts or reference images, we discover two disentanglement properties: (1) Semantic attributes can be edited through specific directions in a controllable manner; (2) Modifying semantics along one controlling direction does not affect other attributes. These properties enable precise and fine-grained modification of specific semantics while preserving others, using easily identifiable editing directions. As shown in Fig. 1(b), given an input image and its text description, we can identify the controlling direction for a specific attribute (e.g., 'smile') by encoding its corresponding text prompt (e.g., the word 'smile'). This enables precise control over the semantic intensity by manipulating the joint latent embedding along this direction.

These properties enable DiT to serve as a prior knowledge for precise semantic editing. Based on this, we propose a simple yet effective **E**ncode-**I**dentify-**M**anipulate (EIM) framework for zero-shot image editing with precise and fine-grained semantic control, while not requiring additional training or mask annotations. Given an input image and target semantic, we first extract the image's text description using a multi-modal LLM, then encode both image and text into a joint latent embedding. To precisely edit the target semantic, we introduce a Hessian-Score-Distillation-Sampling (HSDS) method that identifies disentangled editing directions guided by the target semantic's text prompt, as well as the desired editing degree. The joint latent embedding is then linearly manipulated along this direction. To evaluate our approach, we propose a **S**emantic **D**isentanglement m**E**tric (SDE) for measuring latent space disentanglement in text-to-image diffusion models. We also introduce ZOPIE (**Z**ero-shot **O**pen-source **P**recise **I**mage **E**diting), a new benchmark for comprehensive evaluation. Both automatic and human evaluations demonstrate our framework's effectiveness in precise image editing tasks.

Our contributions are summarized as follows:

- We systematically studied and modeled the latent embedding space of DiT models, and propose a new metric to quantify its *disentanglement*. Our study for the first time comprehensively uncover

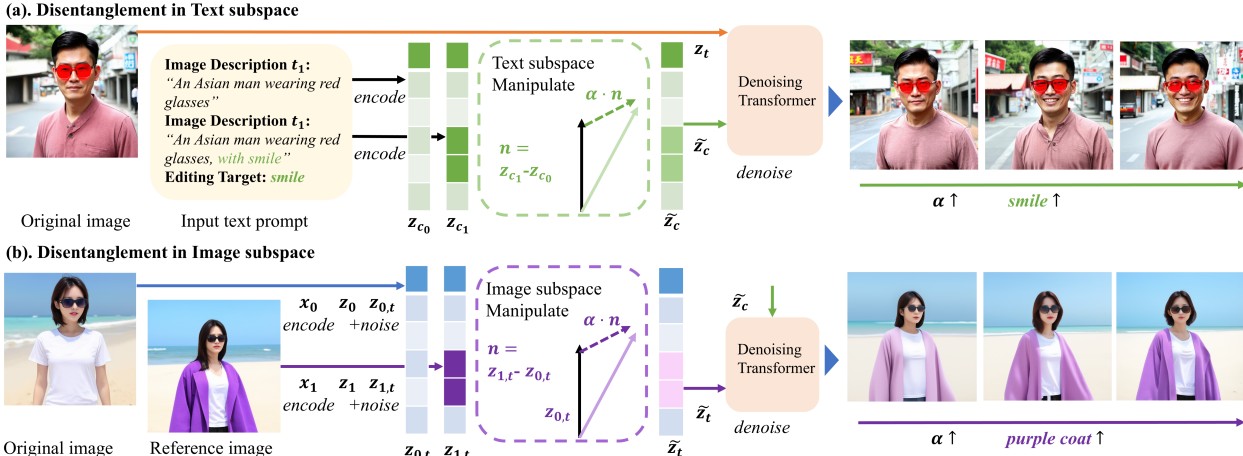

Figure 2: The disentanglement properties of the joint latent space of DiT enable: (a) Given a target semantic to be edited, we can identify a direction that allows fine-grained control of the intensity of this semantic; (b) With a reference image that only differs in the target semantic, we can obtain such direction in the image embedding space and achieve similar precise control.

this important *disentanglement* property of the DiT model, which set the foundation for controllable image editing with DiT models.

- We propose a simple yet effective Extract-Indentify-Manipulation (EIM) method via Hessian Score Distillation Sampling method for zero-shot image editing with precise and fine-grained semantic control, without requiring any additional training or mask annotations.

- We introduce a novel **Z**ero-shot **O**pen-source **P**recise **I**mage **E**diting (ZOPIE) benchmark, with both automatic and human annotations to assess the effectiveness of precise image editing tasks, as well as the disentanglement property of generative models.

## 2 Exploration on Latent Space of Diffusion Transformer

### 2.1 Preliminary

**Diffusion Transformer.** Text-to-Image (T2I) diffusion transformer utilizes (Peebles & Xie, 2023) a transformer model as the denoising backbone. In specific, T2I diffusion transformer employs the encoder of a variational auto-encoder (VAE) model to encode the input image to latent embeddings, a VAE decoder to transform image embeddings back to the pixel space, and a set of text encoders to encode the input text prompt to text embeddings. Given an image and a text prompt, the text encoder will project the prompt into text embeddings $z_c \in \mathbb{R}^{l \times d}$, where $l$ is the length of tokens in the text prompt, and $d$ is the hidden dimension. By averaging $z_c$ across the token dimension, we can obtain a pooled text embedding $\bar{z}_c \in \mathbb{R}^d$ which indicates coarse-grained global semantics. Meanwhile, the image will be encoded into image embeddings $z \in \mathbb{R}^{v \times d}$, where $v$ is the length of the patchified image tokens, and $d$ denotes the hidden dimension, which is the same as text latent embedding. At forward time-step $t$, $z$ will be added with noise $\epsilon_t$, forming noised image embeddings $z_t$. Before the de-noising loops, $z_t$ and $z_c$ are combined into a joint latent embedding and input into the de-noising transformer. Here, the image and text latent embedding dimension are chosen to be the same in order to project both modalities into the joint latent space. They are concatenated to obtain $z \in \mathbb{R}^{(v+l) \times d}$, creating a joint latent space $\mathcal{Z}$. Therefore, this joint latent space consists of two subspaces: the image latent space $\mathcal{Z}_t$, formed by $z_t$, and the text latent space $\mathcal{Z}_c$, formed by $z_c$. After de-noising loops, the model is expected to generate the latent embedding for the edited image, with the VAE decoder will decode the denoised image embedding to pixel space to get the edited image.

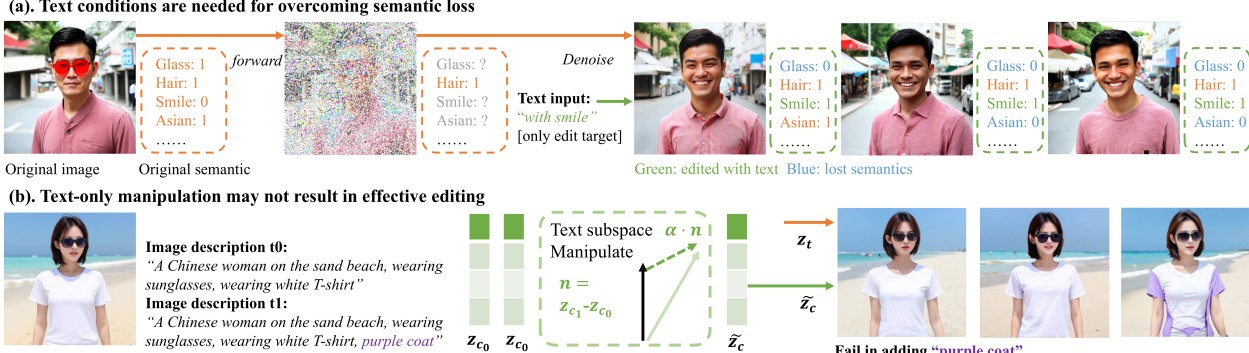

Figure 3: To effectively modify semantics, we must manipulate the entire joint latent space. This is due to two observed challenges: (a) *semantic loss*, where certain semantics are randomly recovered in the denoising process if they are not conditioned; (b) manipulating only text embeddings is insufficient for modifying certain semantics effectively.

**Image Editing**. The investigation and modeling of the disentanglement properties in DiT's latent space are mainly based on the image editing task. The goal of image editing is to modify certain semantic contents in the given source image $x$, while keeping other semantic attributes unchanged. In this paper, we primarily focus on text-guided image editing, as we often edit images based on text instructions in practical applications. Following Wu et al. (2023a), we model the image editing as the two following processes in the latent space. (1) Image forward process: given image embedding $z$ and timestep $t$, $z$ is added with noise $\epsilon_t$ to get noised image embedding $z_t$. (2) Image reverse process: we input text prompts to text encoders to gain text embeddings, and pass them together with $z_t$ into the de-noising transformer to get a de-noised image latent $\tilde{z}$. Then we decode the $\tilde{z}$ to obtain the edited target image $\tilde{x}$. In this paper, to demonstrate the significance of the disentanglement property for controllable image generation, we apply frozen diffusion models to precisely edit targeted semantics. To formalize the goal of this paper: given a semantic $s$ and a source image $x$, *we aim to change the intensity $\alpha$ of a given semantic $s$ reflected in the edited target image $\tilde{x}$, while avoiding incorrectly changing other semantics.* The key differences to the most related works are summarized in Table 4.

## 2.2 Disentangled Semantic Representation Space of Diffusion Transformer

We present our key findings on DiT's latent space, highlighting its unique disentanglement properties compared to related works (Table 4). We illustrate these properties through specific editing examples and our experiments demonstrate their consistency across various cases.

Suppose we are given a source image, a target semantic to be edited, and a text prompt $t_0$ that describes the image. The prompt $t_0$ will be encoder to text embedding $z_{c_0}$, and the source image will be transformed to image embedding $z_t$, both of them are combined to form a joint latent embedding. This joint embedding serves as the input to the de-noising transformer and is straightforward to obtain and manipulate. In the following part, we will demonstrate the semantic disentanglement properties of this joint latent space, as well as how they can be leveraged for editing intensities of specific image semantics.

**Image semantics can be manipulated separately by image or text in the joint latent space.** For a specific semantic content, we first would like to find out the controllable editing direction $n$ in $\mathcal{Z}_c$, which can precisely edit the intensity of this semantic reflected in the image, while leaving the others unchanged. Here, $n$ can be determined explicitly by encoding related text prompts. Specifically, given text prompt $t_1$ that describes the demanded image, and text prompt $t_0$ that describes the original image, we first encode them to obtain text embedding $z_{c_1}$ and $z_{c_0}$ respectively. We then subtract them to obtain $n$. With such direction $n$, given an editing degree $\alpha$, we suppose to manipulate $z_{c_0}$ by applying $\tilde{z}_c = z_{c_0} + \alpha \cdot n$. As shown in Fig. 2, this approach has successfully modified the intensity of the semantic "smile" with fine granularities while avoiding affecting other semantics.

Figure 4: (a) Image-text cross-attention mechanism in UNet-based diffusion models. (b) Self-attention mechanism in the diffusion transformer, where image and text embeddings are concatenated and jointly flowed into the attention block. In the self-attention mechanism, the attention map of a specific semantic contains less category information of other semantics, as shown in Sec. 5.

We have observed similar phenomenons in the image latent space. Similar to the text side, we start from seeking the editing direction $n$. As shown in Fig. 2, we carefully select two images $x_1, x_0$ that are mainly different in the semantic attribute "coat". We encode both of them to obtain image embedding $z_1$ and $z_0$, then we add noise $\epsilon_t$ to them to obtain $z_{1,t}$ and $z_{0,t}$. We found that the distance between $z_{1,t}$ and $z_{0,t}$ can be utilized as the editing direction of the semantic "coat". Therefore, we suppose the editing direction $n$ can be given by $z_{1,t} - z_{0,t}$. We linearly manipulate the image embeddings following $\tilde{z}_t = \alpha \cdot n + z_{0,t}$, where the image embedding $z_{0,t}$ of the image without "coat" shifted toward desired $z_{1,t}$ with different ratio $\alpha$. As $\alpha$ increases, the "coat" semantic gets more significant.

The observations discussed above indicate that image semantics might be encoded into different subspaces, where their intensities are controlled by separate directions. Meanwhile, these directions *can be straightforwardly identified*, either by encoding related text prompts or projecting reference images, if a reference image exists. This disentanglement property may come from the nature that attention maps of semantics in the diffusion transformer are less entangled, as shown in Fig. 4. These findings may unlock the potential of adjusting the intensity of a given semantic, through linearly manipulating latent embeddings near the boundary defined by the editing direction. Detailed theoretical analysis can be found in Appendix A.2.

**Achieving effective semantic editing across different targets requires utilizing the entire joint latent space.** While semantic attributes are encoded in the joint latent space through image and text latent embedding seperately as described in the last section, neither modifications on $\mathcal{Z}_c$ nor $\mathcal{Z}_t$ alone can effectively modify all desired semantics in a precise control manner. Firstly, during the forward process, $\mathcal{Z}_t$ may suffer from the 'Semantic Loss Phenomenon' Yue et al. (2024), losing some of the semantics $\mathcal{Z}_0$ contained. As demonstrated in Fig. 3, in the forward process, some attributes become noised; they will be assigned random values if not conditioned by text embeddings during the reverse process. For example, in Fig. 3, when the semantics 'ethnicity' and 'glasses' are left unconditioned on the text side, the reconstructed image samples exhibit random race and glasses attributes. In contrast, when conditioned by text embeddings, the attributes are correctly recovered, as shown in Fig. 2. This observation indicates that some semantics cannot be effectively conditioned if text embeddings are not well leveraged, suggesting that $\mathcal{Z}_t$ alone is insufficient for effective control.

Conversely, manipulating only $\mathcal{Z}_c$ fails to edit certain semantics. Fig. 3 demonstrates this limitation when attempting to modify a 'no coat' image to include a 'purple coat', where we can see by only editing the text prompt while not manipulate the image embedding, the model fails to add a correct 'purple coat' as desried. As shown in Fig. 2(b), the edit succeeds only when we manipulate both the image and text latent spaces by reweighting $z_t$ of the source image along with reference images containing purple coats. Here we assume the access to the reference image for the purpose of analysis. However, the reference image, which is actually the editing target image, does not exist in practical image editing tasks. In the following Method section, we will discuss how to manipulate $z_t$ without reference images.

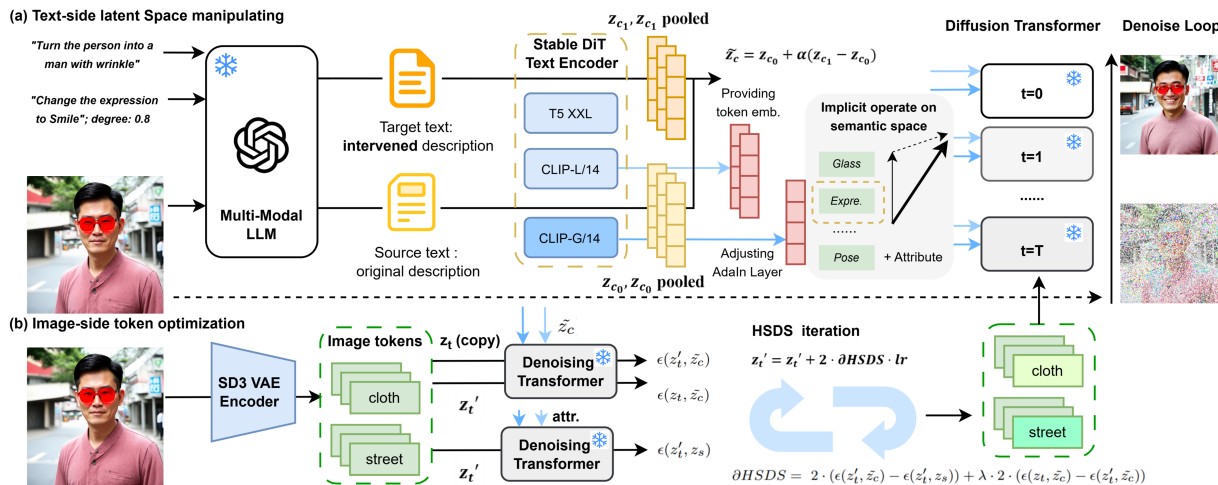

Figure 5: (a). **Encode**: Given a source image, we first utilize a multi-modal LLM to get a source prompt that describes semantics of the image, and a target prompt for the desired image. We encode the source prompt and the given image to a joint latent embedding. (b). **Identify**: We obtain the editing direction in text subspace by subtracting embeddings encoded from the source prompt and target prompt. We propose a Hessian-Score-Distillation-Sampling method to identify the editing direction in image subspace. (c). **Manipulate.** Finally, we linearly combine the joint latent embedding with on the identified direction.

## 3 Method

### 3.1 Encode-Identify-Manipulate Framework

Motivated by the disentanglement properties of diffusion transformers, we propose our **E**ncode-**I**dentify-**M**anipulate (EIM) framework for precise image editing (Fig. 5). Given a source image $x$ and target semantic $s$ (e.g., 'smile'), our goal is to precisely control the intensity of $s$ in the edited image.

The framework requires both the source image's text description $t_0$ and the desired image's description $t_1$ to form the joint latent embedding, leveraging DiT's image-text self-attention mechanism. When these descriptions are unavailable, we generate them using a multi-modal LLM. [2].

**Encode.** As demonstrated in Sec 2.2, we have shown that when the task-irrelevant semantics are not conditioned during the denoising process, they will be randomly changed due to the semantic loss phenomenon. Thus, we need to inject the text embedding $z_{c_0}$ of text prompt $t_0$, which describes the source image, as the condition to help correctly maintain these semantics. Meanwhile, the given image $x$ is encoded into image embedding $z$, and is added with a randomly sampled noise $\epsilon_t$ at timestep $t$, to obtain the noised image embedding $z_t$. And we also encode the text description (e.g., 'purple coat') of our editing target to a text embedding $z_s$.

**Identify.** In order to manipulate the joint latent embedding for editing target image semantics, we aim to identify an editing direction that controls this semantic. Specifically, since $\mathcal{Z}$ consists of subspaces $\mathcal{Z}_t$ and $\mathcal{Z}_c$, we can separately obtain the editing direction for each subspace. For text latent space, the editing directions are explicitly associated with text prompts. Therefore, given the ratio $\alpha$ for modifying the semantic $s$, we can firstly encode the text $t_1$ which describes the desired image, to obtain text embedding $z_{c_1}$. Then we construct the editing direction $n$ here by $n_c = \alpha(z_{c_1} - z_{c_0})$. Therefore, we can operate the text embedding with: $\tilde{z}_c = z_{c_0} + n_c$.

In practice, for the noised image embeddings $z_t$, assuming that there is no reference image, we lack an explicit editing direction as in the text side. To address this problem, we obtain this direction based on a

---

[2]In our method, we instruct the multi-modal LLM to generate text descriptions for both the original and desired images, ensuring that they differ only in the tokens corresponding to the semantics to be edited. Consequently, in the encoded text embeddings, only parts related to the specified semantics are altered.

score distillation sampling (Hertz et al., 2023; Poole et al., 2022) method, which computes an update direction $n_{z_t}$ for latent image embedding $z_t$. During the de-noising process, we have observed that de-noising with $z_s$ that encoded from the text prompt of the desired semantic (e.g., 'purple coat'), leads to higher intensity of this semantic in the generated image. Thus, we assume the difference of the two predicted noise conditioned on $z_s$ and $z_{c_0}$, can offer a rough direction for modifying the intensity of this semantic. Therefore, it is straightforward to employ delta distillation sampling (Hertz et al., 2023) to push image embedding $z_t$ to $z'_t$ along the rough editing direction, which follows $\partial(\|\epsilon(z'_t, z_s) - \epsilon_t\|_2 - \|\epsilon(z_t, z_{c_0}) - \epsilon_t\|_2)$ as the update direction. Here $\epsilon_t$ is a random noise sampled at time-step[3] $t$.

However, this might incorrectly affect the semantics controlled by text embeddings. In contrast, we should encourage the image embedding moves in a precise direction, to avoid affecting semantics irrelevant to the editing task. Motivated by this insight, we propose a Hessian Score Distillation Sampling (HSDS) method, which aims to minimize the distance between $\epsilon(z'_t, \tilde{z_c}) - \epsilon(z_t, z_{c_0})$ and $\epsilon(z'_t, z_s) - \epsilon(z_t, z_{c_0})$. This aligns $\epsilon(z'_t, \tilde{z_c})$ and $\epsilon(z'_t, z_s)$ alongside the direction that can increase the intensity of the target semantic, while mitigating unintended changes on other semantics. Meanwhile, we add a constrained term which aims to reduce the distance between $\epsilon(z'_t, \tilde{z_c})$ and noise $\epsilon(z_t, \tilde{z_c})$ predicted using $z_t$ and $\tilde{c}$ as input. Finally, the image gradient of the Hessian Score-Distillation-Sampling should be:

$$\partial HSDS = 2 \cdot (\epsilon(z'_t, \tilde{z_c}) - \epsilon(z'_t, z_s)) + \lambda \cdot 2 \cdot (\epsilon(z_t, \tilde{z_c}) - \epsilon(z'_t, \tilde{z_c})) \tag{1}$$

, where $\lambda$ is the hyper-parameter that adjusts the degree to which the manipulated image embedding $z'_t$ should be close to the original $z_t$. Detailed analysis can be seen in appendix A.4.

In score-distillation-sampling, we do not require the gradient of the diffusion transformer. Hence we can directly compute $\partial HSDS$, and update the image embedding with: $z'_t = z_t + n_{z_t}, n_{z_t} = \eta \cdot \partial HSDS$, where $\eta$ refers to the step size for the update loop. For implementation, we iteratively update $z_t$ to obtain a more accurate $n_{z_t}$.

**Manipulate.** We manipulate the joint latent embedding through separate linear combinations in the text and image subspaces, with the identified editing direction $n_c$ and $n_{z_t}$. For the text side, we operate the text embedding with: $\tilde{z_c} = z_{c_0} + n_c$. Similarly, we operate $z_t$ with $\tilde{z_t} = z_t + n_{z_t}$.

## 3.2 Semantic Disentanglement Metric

Based on disentanglement properties demonstrated in Sec. 2, we propose a metric that can be automatically computed to measure the degree of disentanglement of the latent space of text-to-image diffusion models. As we mentioned in Sec. 2.2, a disentangled semantic representation space $S'$ should satisfy decomposability and effectiveness properties, where we can locally manipulate certain subspaces to effectively modify specific semantics without altering other semantics. Building on this insight, we propose to evaluate the disentanglement degree of a semantic representation space using an image-editing-based approach. Specifically, we aim to examine the following properties[4]: (1). **Effectiveness**: Can we effectively edit certain semantics in the latent space? (2). **Decomposable**: Manipulating latent embedding along editing direction of a specific semantic, would not affect the other semantics?

Consider an image $x$ with a binary semantic $s \in [0, 1]$ that will be lost in the forward process, which is labeled as $y_0 = 0$. We encode $x$ to latent space to get image embedding $z$, and forward $z$ to obtain $z_t$. We then denoise $z_t$ in the reverse process, and decode it to get the reconstructed image. To ensure effective editing, the reconstructed image $\tilde{x}$, conditioned on $y_1 = 1$, should differ significantly from $\hat{x}$, which is recovered under the condition $y_0 = 0$. This difference is measured by comparing their respective distances to the original $x$. Simultaneously, to maintain the decomposability property, the distance between $x$ and $\tilde{x}$ should be small, ensuring that task-irrelevant semantics remain unchanged during the editing process.

*Remark* 1. Given a diffusion model with a encode $f$ which turns a given image into the noised image embedding, and a decode function $h$ which turns noised image embedding to the recovered image, an image labeled with $y \in [0, 1]$ of semantics $s$, and text embeddings $c, \tilde{c}$ encoded from text prompts correspond to $y$

---

[3]Different from SDS that focus on optimizing image embedding $z$, we aim to utilize SDS-based method to seek an editing direction for noised image embedding $z_t$ at timestep $t$, thus we hold a fixed time-step $t$ when utilizing SDS-based method.

[4]Detailed modeling and analysis of these properties can be seen in the appendix A.1.

and $1 - y$. The **S**emantic **D**isentanglement m**E**tric (SDE) could be written as:

$$\text{SDE} = \frac{||x - h(f(x,t), c, t)||_2}{||x - h(f(x,t), \tilde{c}, t)||_2} + ||x - h(f(x,t), \tilde{c}, t)||_2 \qquad (2)$$

where $t$ is the forward time-step.

## 4 Experiment

### 4.1 Experiment Setting

**Implementation Detail**. We include three state-of-the-art open-source transformer-based Text-to-Image (T2I) diffusion model for our experiments: Stable Diffusion V3 Esser et al. (2024), Stable Diffusion V3.5[5], and Flux[6]. We follow their implementation in the Huggingface Diffusers platform von Platen et al. (2022). During image editing tasks, we set the classifier-free-guidance (CFG) scale to 7.5 and the total sampling steps to 50. We forward the image to 75% of the total timesteps, and then conduct the reverse process. For UNet-based T2I model, we include the widely utilized Stable Diffusion V2.1 from Huggingface Diffusers. We set the hyperparameters in the same way as those mentioned above. During our experiments, we keep diffusion models frozen. We utilize the GPT-4 Achiam et al. (2023) to extract text descriptions of given images. Details can be found in the appendix C.2 and C.1.

**Baselines.** Text-guided zero-shot precise image editing is a novel task; we adapt and compare to existing text-guided image editing methods DiffEdit Couairon et al. (2022), Pix2Pix Brooks et al. (2023), and MasaCtrl Cao et al. (2023) as our baselines. To enable precise image editing, we apply similar text manipulation operations to the text embeddings that guide the reverse process in image editing. We also include baseline methods that utilize the diffusion transformer but modify text embeddings only or manipulate image embeddings only.

**Qualitative Evaluation.** We conduct a comprehensive qualitative assessment of EIM using a diverse array of real images spanning multiple domains. Our evaluation employ simple text prompts to describe various editing categories, including but not limited to style, appearance, shape, texture, color, and lighting. These edits are applied to a wide range of objects, such as humans, animals, landscapes, vehicles, and surfaces. To demonstrate the necessity of image-side manipulation, we compare EIM's performance with text-only embedding space manipulations. Table 1 shows a checklist of the qualitatively investigated image editing tasks along with illustrative examples. For our source images, we generate high-resolution samples using diffusion transformers. We then apply our method with three different editing degrees to showcase precise editing capabilities and the controllability of the editing degree.

Additionally, we compare the precise editing results of our EIM method with those of the baseline models. We adapt the baselines to the same experimental pipeline for precise image editing. Like EIM, these methods are applied with three different editing degrees to demonstrate their capabilities.

**Quantitative Evaluation.** To address the lack of evaluation benchmarks for precise zero-shot image editing, we introduce a Zero-shot Open-source Precise Image Editing (ZOPIE) benchmark. ZOPIE includes both human subjective and automatic objective evaluations, focusing on image quality, controllability, image-text consistency, background preservation, and semantic disentanglement. The benchmark consists of 576 images across 8 editing types and 8 object categories, with annotations for source prompt, edit prompt, editing feature, object class, region of interest, and background mask. Details are provided in the appendix C.1.

For quantitative evaluation, we employed six metrics across four main aspects: 1) Background preservation: PSNR, LPIPS, and SSIM. 2) Text-image consistency: CLIPScore. 3) Editing effectiveness: Multimodal LLM-based Visual Question Answering score (MLLM-VQA score), which leverages multimodal language models to assess the smoothness and perceptibility of semantic intensity control.

The CLIPS score is calculated between the edit prompt and the edited images. For the MLLM-VQA score, pairs of images with consecutive editing strengths are fed into GPT-4 with the question: "Answer only in

---

[5]https://huggingface.co/stabilityai/stable-diffusion-3.5-medium
[6]https://github.com/black-forest-labs/flux

yes or no, does the second image reflect a gradual change of [edit feature] compared to the first image?" This process is repeated five times per sample, with the percentage of "yes" responses used to quantify the effectiveness of precise editing. Evaluation details can be seen in the appendix C.1.

**Evaluation on semantic disentanglement degree.** We use the SDE metric proposed in Sec. 3.2 to assess the disentanglement degree of text-to-image diffusion models quantitatively. Specifically, we calculate the SDE for both the UNet-based diffusion model and the diffusion transformer using the CeleBA Liu et al. (2015) dataset—a widely used benchmark for evaluating generative capabilities in diffusion models Rombach et al. (2022); Zhang et al. (2024a). Each image in CeleBA is associated with an attribute vector, making it well-suited for measuring disentanglement with the SDE metric. We randomly sample 2,000 images and evaluated the models' disentanglement degree across age, gender, expression, hair, eyeglasses, and hat attributes.

Table 1: Checklist of the qualitatively investigated editing task. Text tokens are more efficient in controlling semantics related to person and object than attribute and texture.

| method | Attribute | | | Texture | | | textbfPerson | | | Object | | |
|---|---|---|---|---|---|---|---|---|---|---|---|---|
| | Category | Example | | Category | Example | | Category | Example | | Category | Example | |
| EIM | quantity | tiny→huge | ✓ | light | bright→dark | ✓ | smile | slight→intense | ✓ | cover | none→coat | ✓ |
| EIM | shape | rect.→spher. | ✓ | surface | uneven→iron | ✓ | age | young→old | ✓ | small item | pig→cat | ✓ |
| EIM | color | blue → green | ✓ | style | pencil→painting | ✓ | wrinkle | severe→slight | ✓ | large object | kangaroo→bear | ✓ |
| Text-only | quantity | tiny→huge | | light | bright→dark | | smile | slight→intense | ✓ | cover | none→coat | |
| Text-only | shape | rect.→spher. | | surface | uneven→iron | ✓ | age | young→old | ✓ | small item | pig→cat | ✓ |
| Text-only | color | blue → green | ✓ | style | pencil→painting | | wrinkle | severe→slight | ✓ | large object | kangaroo→bear | ✓ |

## 4.2 Qualitative Results

**EIM achieves greater effectiveness in editing target semantics by operating on the entire semantic representation space.** Fig.6 compares the images resulting from manipulating the entire joint latent embedding versus those from manipulating only the text embedding. While our EIM method can effectively edit target semantics by modifying the whole latent embedding, there are some failure cases in employing only text-side manipulation. For example, for editing tasks "enlarging a cat sitting in the garden" and "adding a purple coat to an American woman by the sea," the image subspace should also be manipulated for effective image semantic editing. More examples can be seen in the appendix E.

**EIM outperforms baseline approaches in precise image editing and controls the intensity of target semantics in fine granularities.** As illustrated in Fig. 8, the baseline methods frequently introduce unintended changes to irrelevant semantics, disrupting the overall image coherence. In contrast, our EIM method precisely modifies target semantics while keeping others unchanged, allowing for fine-grained adjustments to the intensity of the target semantics. More examples can be seen in the appendix E.

## 4.3 Quantitative Results

**Our EIM method achieves better precise image editing results.** As shown in Table 2, EIM demonstrates a significant advantage in background preservation during edits, with a strong PSNR of 20.8 and the lowest LPIPS score of 133.6. This showcases EIM's ability to decompose and isolate specific semantics, ensuring that areas irrelevant to the editing target remain unaffected by the modifications. Moreover, the high CLIP Similarity Score of 34.3 achieved by EIM highlights its effectiveness in accurately editing target semantics. Finally, the significant advantage observed in the MLLM-VQA metric, where EIM scores 43.8, demonstrates its strength in fine-grained editing, reflecting the model's ability to produce nuanced and precise modifications.

**The diffusion transformer has better semantic disentanglement degree.** As indicated in Table 3, the diffusion transformer consistently achieves lower SDE scores across various attributes such as age, gender, expression, hair, eyeglasses, and hat, compared to the UNet-based diffusion model. For example, the average SDE score of the stable diffusion 3 is significantly better compared to that of the UNet-based model. This suggests that the diffusion transformer has a more disentangled representation space, leading to more precise control during image editing.

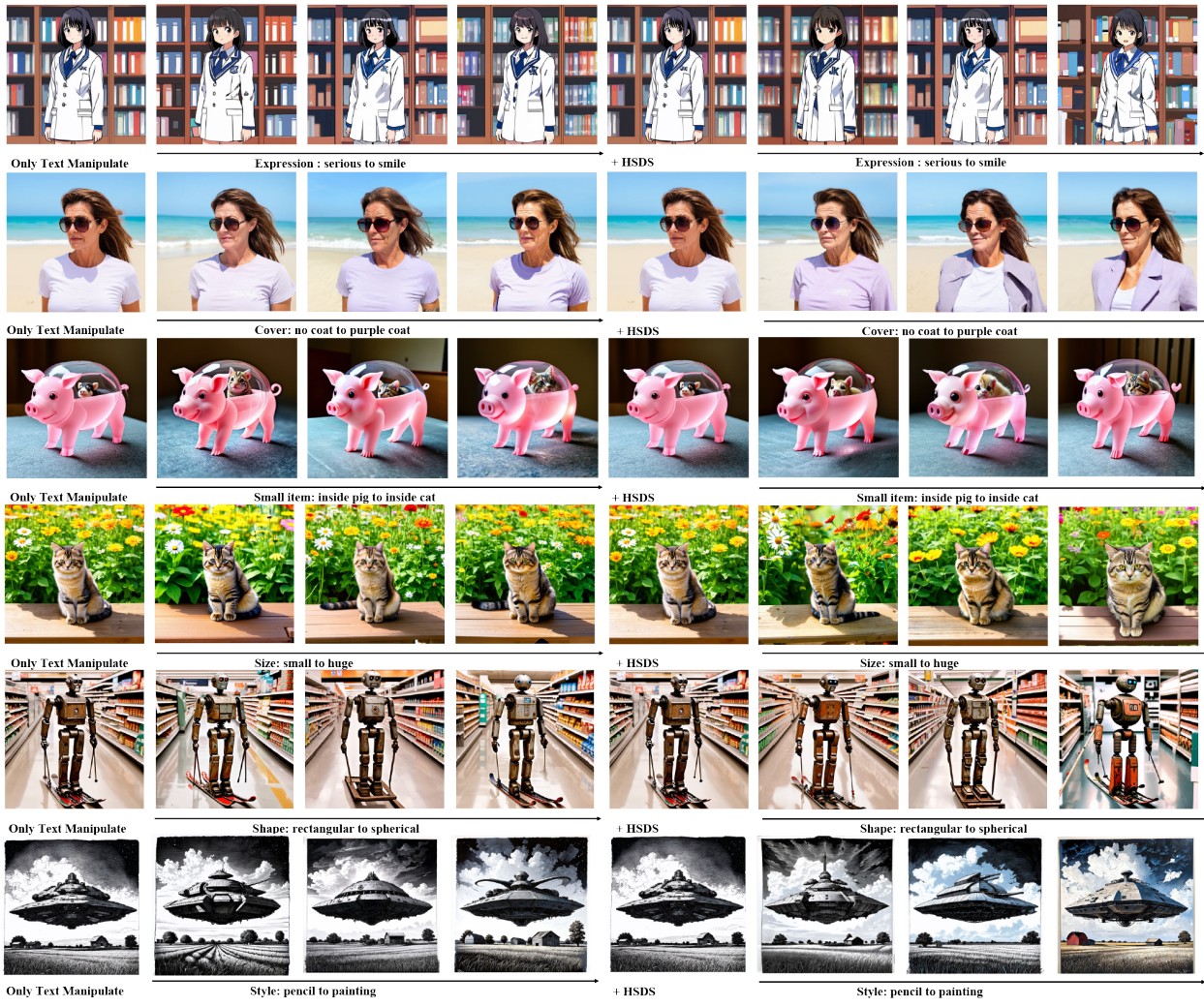

Figure 6: Example precise image editing results of diverse semantics such as texture, shape, size, and expression. We utilize diffusion transformers to modify the intensity of considered attributes in fine granularity. The results on the left are from text-only manipulation, and on the right are from the complete EIM method.

Table 2: Quantitative Results

| Method | Background Preservation | | | CLIP Similarity Score | MLLM-VQA |
|---|---|---|---|---|---|
| | PSNR ↑ | LPIPS ↓ ×$10^3$ | SSIM ↑ ×$10^2$ | Whole ↑ ×$10^2$ | Whole ↑ ×$10^2$ |
| P2P Brooks et al. (2023) | **21.0** | 139 | 79.5 | 30.1 | 16.7 |
| DiffEdit Couairon et al. (2022) | 19.1 | 205.4 | 69.9 | 30.7 | 14.6 |
| MasaCtrl Cao et al. (2023) | 16.4 | 195.8 | 68.7 | 28.5 | 18.5 |
| Ours (Prompt Only) | 20.2 | 183.5 | 73.9 | 31.2 | 29.2 |
| Ours (HSDS Only) | 18.9 | 190.2 | 71.5 | 30.9 | 27.4 |
| Ours | 20.8 | **133.6** | **79.6** | **34.3** | **43.8** |

| Model | Backbone | SDE on Age↓ | SDE on Gender↓ | SDE on Expression↓ | SDE on Hair↓ | SDE on Eyeglasses↓ | SDE on Hat↓ |
|-------|----------|-------------|-----------------|---------------------|---------------|---------------------|--------------|
| SD2 Rombach et al. (2022) | UNet | 1.42 | 1.12 | 1.42 | 1.29 | 1.28 | 1.27 |
| SD3 Esser et al. (2024) | Transformer | 1.14 | 1.08 | 1.15 | 1.19 | **1.12** | 1.10 |
| SD3.5 | Transformer | 1.11 | 1.12 | **1.10** | **1.18** | 1.13 | **1.07** |
| Flux | Transformer | **1.09** | **1.07** | 1.13 | 1.21 | 1.15 | 1.08 |

Table 3: Semantic disentanglement metric of the diffusion transformer and the UNet-based diffusion model.

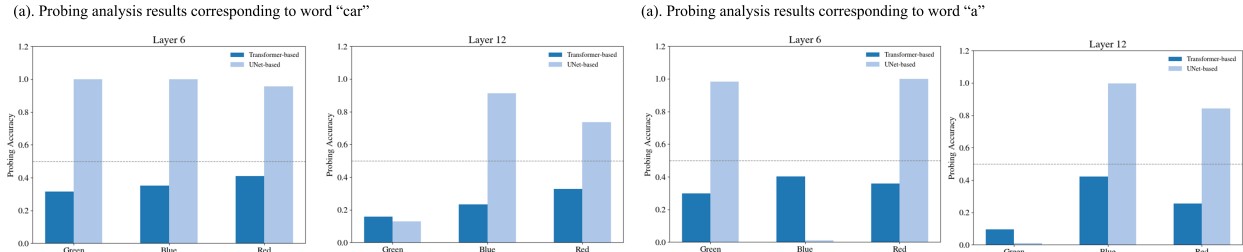

Figure 7: Results of the probing analysis. We provide ratios of attention maps from different models being classified into various color categories. when tested against corresponding color classifiers.

## 5 Analysis Study

### 5.1 Probing Analysis on Disentanglement Mechanism

In this section, we will demonstrate the potential origin of the disentanglement properties of diffusion transformer's latent space, as well as why the representations of classical UNet-based diffusion models are always entangled.

We start by revisiting different text conditioning approaches of these two types of models. As shown in Fig 4 (a), the denoising block of the diffusion transformer utilizes a self-attention mechanism Esser et al. (2024) that early concatenates image and text embeddings, allowing them to be jointly flowed into self-attention blocks to predict the noise. In contrast, as demonstrated in Fig 4 (b), the UNet architecture employs cross-attention layers to introduce text conditions into the denoising process, which might learn entangled mapping from input conditions to image semantics Liu et al. (2024). For example, when generating images with UNet-based diffusion models using the caption $"a <color> <object>"$, the attention map w.r.t. semantic "<color>" might contain category information of semantic "<object>". Therefore, when we aim to edit semantic "<color>", the semantic "<object>" would be incorrectly modified.

Building on this insight, to understand the origin of the disentanglement effect, we conduct a probing analysis Clark (2019); Liu et al. (2019) to explore the differences between the self-attention and cross-attention mechanisms in guiding image generation, as utilized in transformer-based and UNet-based diffusion models, respectively. Specifically, we generate images using the caption "a <color> <object>" and analyze the attention maps corresponding to each token in the intermediate layers. We then train classifiers using the attention maps of "<color>" and apply these classifiers to the attention maps of "<object>. If classifiers successfully identify color information within the "<object>" attention maps, the attention map may contain category information related to "<color>".

**The attention map for a specific semantic in the diffusion transformer does not contain category information from other semantics.** As illustrated in Fig. 7, the ratio of the diffusion transformer's attention map for "<car>" being classified with a specific color is near 0.5, indicating that it does not effectively encode color category information. This contrasts with the attention map of "<car>" in UNet-based diffusion models, which shows a higher tendency in identifying color categories. This observation suggests that the disentanglement property in diffusion transformers stems from their image-text joint self-attention mechanism, leading to less entanglement in the representations corresponding to each semantic. Which similar to the findings in Zhang et al. (2024b). Further experimental details and results are provided in Appendix F.

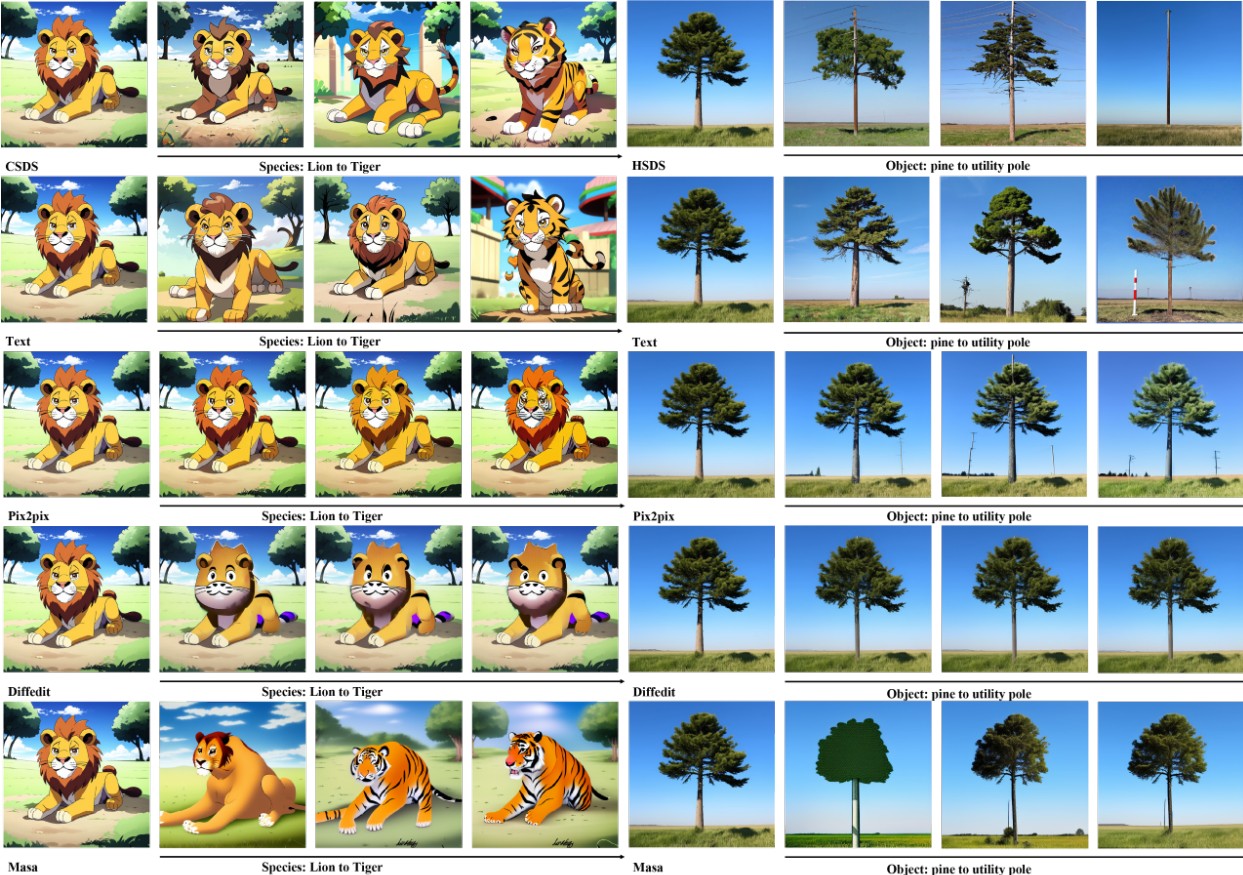

Figure 8: Comparisons of fine-grained editing results of our EIM method and baseline methods. Our EIM method has successfully achieved progressive transformations from lion to tiger and pine tree to utility pole. While baseline methods fail to achieve the target transformations, often degrading the original image quality significantly or producing unintended effects.

## 5.2 Bidirectional Editing and Limitations

When the latent vector moves across the hyperplane formed by the editing direction based on the target value of the semantic (e.g., semantic value "smile" of the semantic "expression", as discussed in Sec. 2.2), the sign of the distance between the latent representation and the hyperplane flips. This allows us to both increase and decrease the intensity of a given semantic value.

Formally, given a normal vector $n_i$ of editing direction for semantic $s_i, i \in [1, ..., m], m \in N^+$, and a latent representation $z$, the signed distance to the hyperplane is given by: $D = n_i^T z$. As $z$ moves in the direction of $n_i$ or $-n_i$, the distance $d$ becomes positive or negative, respectively, allowing bidirectional control of the attribute.

As shown in Fig. 9, we have achieved bi-directional image editing across the boundary defined by the editing direction. We first manipulate the latent embedding along the direction given by our method to obtain the edited image. Then, the latent embedding is manipulated in the opposite direction to obtain the semantic reversed image. As shown in the figure, the expression of the woman is first changed to "smile" and then changed from "smile" to "serious".

We also find that effective manipulations with $n$ should be near the boundary defined by it. In other words, the editing degree $\alpha$ could not be extremely large. As demonstrated in Fig. 9, when we move the latent vector $z$ too far from the hyperplane, exceeding a certain threshold, the probability of the projection falling near the hyperplane diminishes rapidly. This is consistent with the concentration of measure phenomenon

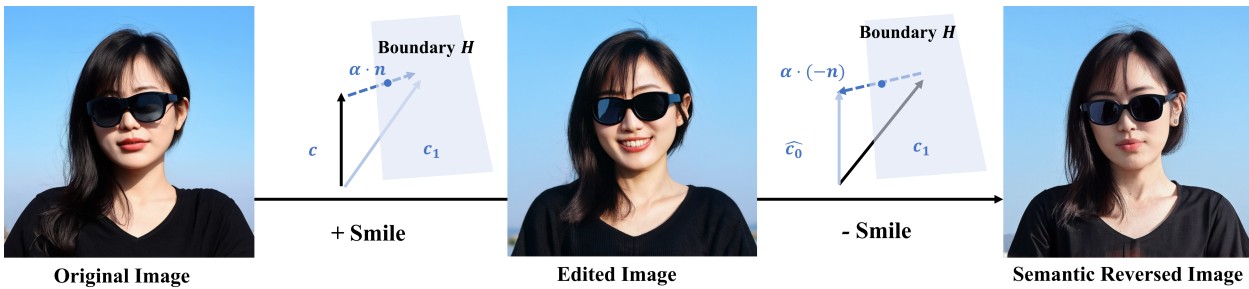

Figure 9: Edited semantics can be reverted to their original values. The disentanglement property allows precise editing of the target semantic without affecting others, enabling us to use the same direction that was used to edit the semantic to reverse it back to its original value.

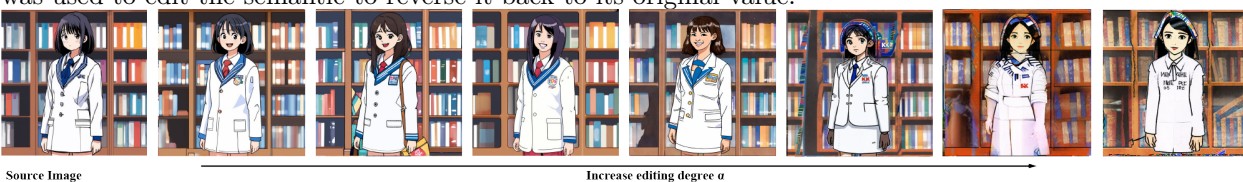

Figure 10: Linear manipulation is effective near the boundary defined by the editing direction. However, if the edit degree exceeds the threshold set by Proposition.(1), the resulting image may become corrupted.

in high-dimensional spaces Shen et al. (2020). In our work, the threshold can be given by Proposition 1 in Sec. 2.2. When the threshold is exceeded, we observe a loss of controlled attribute editing. For example, in our experiments with facial attributes, we find that extreme modifications to the "expression" semantic begin to affect unrelated features such as ethnicity, as shown in Fig. 10.

## 5.3 Hyperparameter Analysis

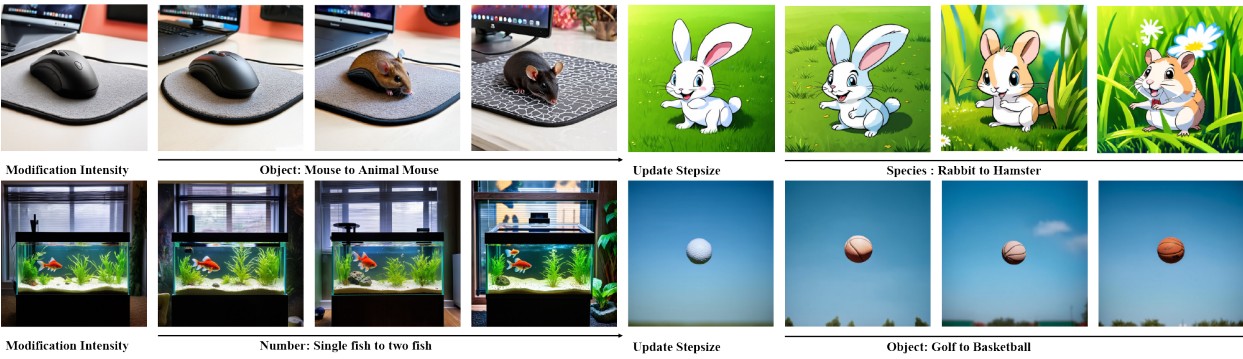

Figure 11: Ablation study on hyperparameters in our EIM method. The figure demonstrates the effects of modification intensity ($\lambda$) and update stepsize on various editing tasks. These examples highlight how careful tuning of these hyperparameters allows for precise control over edit strength, content preservation, and the balance between desired modifications and maintaining original image semantics.

In addition to the editing strength $\alpha$, two crucial hyperparameters in our EIM method are the update stepsize and the modification intensity $\lambda$ used in the Hessian score-distillation sampling (HSDS) process.

**Update Stepsize:** The update stepsize $\eta$ determines the magnitude of each update during the Score-Distillation Sampling process. We find that a carefully chosen stepsize is essential for achieving smooth and controlled edits. Too large a stepsize can lead to overshooting and instability in the editing process, while too small a stepsize can result in slow convergence and subtle edits.

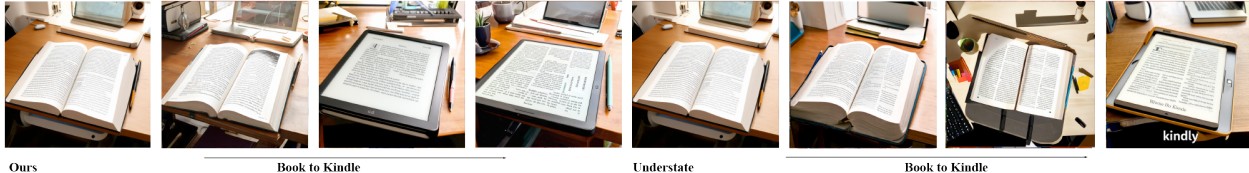

Figure 12: Highlight editing targets in the HSDS can help manipulate image embedding more effectively.

In our experiments, we employ an adaptive stepsize strategy, starting with a relatively large value of 0.1 and gradually decreasing it to 0.01 as it progresses. This approach allows for rapid initial progress followed by finer adjustments, resulting in more precise edits.

**Modification Intensity $\lambda$:** The modification intensity $\lambda$ in our HSDS objective (Eq. 1) plays a crucial role in balancing between achieving the desired edit and preserving the original image content. It controls the trade-off between modifying the target attribute and maintaining other semantic features of the image. We empirically observe that setting $\lambda$ in the range of 0.1 to 1.0 generally produces good results across various editing tasks. Specifically, a higher $\lambda$ values (0.7 - 1.0) result in more conservative edits that better preserve the original image structure but may require more iterations to achieve the desired effect. A Lower $\lambda$ values (0.1 - 0.3) allow for more aggressive edits but may occasionally introduce unintended changes in other attributes. The optimal value of $\lambda$ can vary depending on the specific attribute being edited and the desired strength of the edit. For example, we observe that edits involving more localized changes (e.g. facial features) generally benefit from higher $\lambda$ values, while global edits (e.g. overall style or lighting) can tolerate lower $\lambda$ values. Figure 11 illustrates the effect of different $\lambda$ values on the editing process for a sample image, demonstrating the trade-off between edit strength and content preservation.

**Score-Distillation-Sampling (SDS) guided by text prompt of the editing target can provide a more effective editing direction.** We use the text embedding encoded from the text prompt of the given editing target, to guide the naive SDS process to provide an editing direction. As shown in Fig. 12, compared to utilizing the text description of the source image to guide the SDS process, this approach leads to more effective image manipulation, but results in more unintended changes. Therefore, use text prompt of the editing target in SDS can provide a more effective but less precise editing direction. This finding has motivated our HSDS method as discussed in the appendix A.4.

### 5.4 Detailed Analysis on Semantic Loss

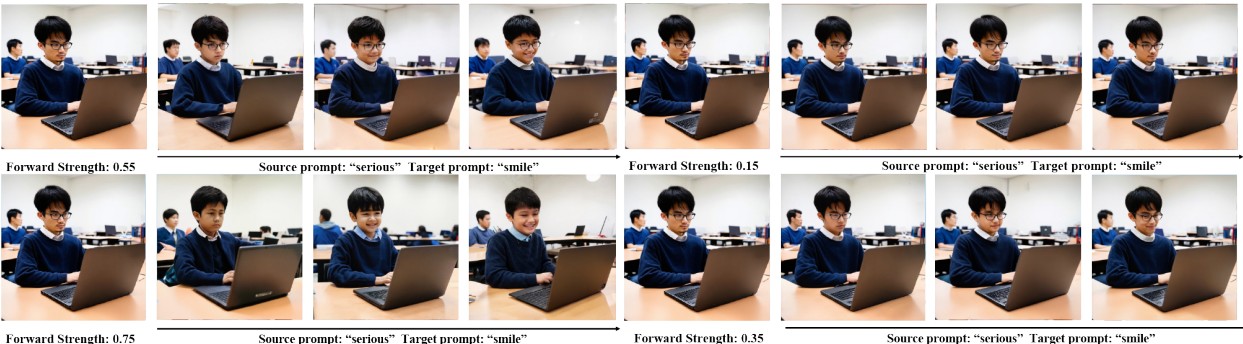

Figure 13: Semantics are gradually lost during the forward process. If not conditioned, these lost semantics are randomly recovered. As the forward strength increases, semantics such as 'age,' 'ethnicity,' and 'glasses' are sequentially lost, and in the recovered images, these semantics are assigned random values.

**In image editing, semantics are gradually lost as the forward strength increases.** We forward the given image with 4 different strengths, resulting in 4 sets of noised images with different noise degrees. We

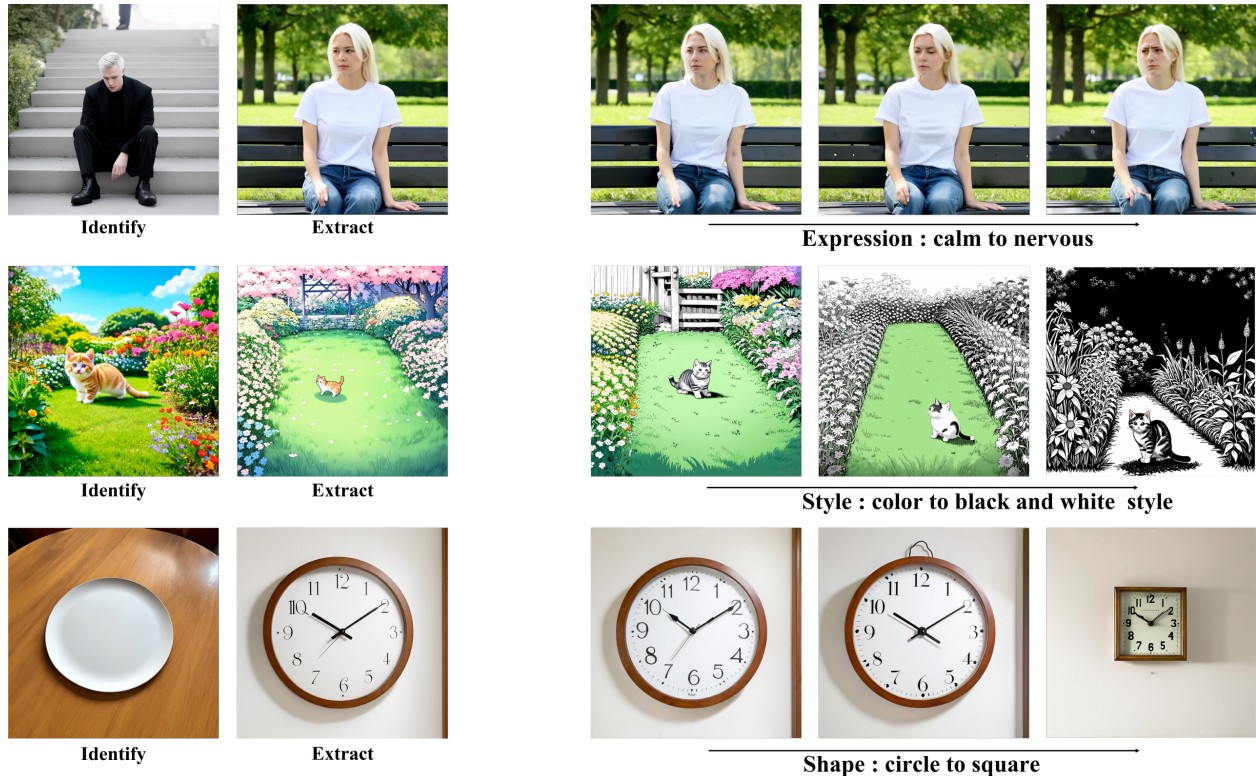

Figure 14: Generalization of the identified editing direction.

de-noise the images conditioned on the editing target, and obtain 4 sets of recovered images. As shown in Fig. 13, at lower forward strengths (e.g., 0.15), the model successfully preserves the target semantic "smile" while keeping other attributes, such as "ethnicity," unchanged. However, the semantic "smile" has not been significantly changed when forward strength is 0.15. Meanwhile, as the forward strength increases to 0.55 and beyond, critical semantics like "ethnicity," "glasses," and "age" are replaced with random values in the reconstructed images. This progression demonstrates the trade-off required in controlling forward strength to prevent unintended alterations in semantics irrelevant to editing tasks.

Table 4: Comparisons between related works and our paper. We organize related works according to our key areas of interest. For properties of the considered semantic latent space, we focus on: (1) Semantic latent space is easy to identify. (2) Image semantics are not entangled. (3) Linear manipulation of the semantic space can linearly modify the intensity of semantics. For the image editing application, we focus on: (1) Precisely edit intensity of semantics. (2) Zero-shot image editing. (3) No need for reference images.

| Method | Easy Identify | Disentangled | Linearity | Precise | Zero-Shot | Reference-Free |
|---|---|---|---|---|---|---|
| Hertz et al. (2022) | | | | | ✓ | |
| Cao et al. (2023) | | | | | ✓ | |
| Brooks et al. (2023) | | | | | | ✓ |
| Couairon et al. (2022) | | | | | ✓ | ✓ |
| Hertz et al. (2023) | | | | | ✓ | ✓ |
| Wu et al. (2023a) | ✓ | | | | | ✓ |
| Kawar et al. (2023) | | ✓ | | ✓ | | ✓ |
| EIM (ours) | ✓ | ✓ | ✓ | ✓ | ✓ | ✓ |

With original image, no editing target

Aggressive Edit          Refinement

With original image, has editing target

Aggressive Edit

Figure 15: Our method can be utilized to refine the aggressively edited images. Here the source image's age semantic was aggressively edited by an off-the-shelf model, and the resulting image can be refined by our method.

## 6 Related Work

### 6.1 Diffusion Model

In recent years, diffusion models have solidified their position as leading techniques in image generation, particularly for producing high-quality and diverse visuals Ho et al. (2020); Rombach et al. (2022); Esser et al. (2024). These models traditionally introduce noise to images in a forward process and utilize a U-Net architecture Ronneberger et al. (2015) to predict the denoised outputs. Text-to-image diffusion models have particularly stood out, surpassing traditional generative models like GANs Karras et al. (2019) in both image quality and zero-shot synthesis capabilities Rombach et al. (2022); Ramesh et al. (2022). While U-Net has been the standard architecture for denoising, recent research has shifted towards transformer-based diffusion models Peebles & Xie (2023); Esser et al. (2024), demonstrating superior performance in generating high-fidelity and diverse images. Despite the successes of diffusion transformers, the precise mechanisms by which input conditions influence the generation process remain largely unexplored.

### 6.2 Semantic Disentanglement in Generative Model

Semantic disentanglement is an essential property in previous generative models, like Generative Adversarial Networks Goodfellow et al. (2020); Karras et al. (2019) and $\beta$-Variational Autoencoder Burgess et al. (2018), enabling precise control over how specific semantics are represented in the generated samples. With a generative model that has a well-disentangled latent space, we can modify the values of specific semantics in the latent space without affecting others, thereby achieving precise and precise image editing Shen et al. (2020). By adjusting specific directions in this latent space, it is possible to modify only the targeted semantics Härkönen et al. (2020). While these models exhibit decent results in continuously and precisely modifying specific semantics, their generated samples are restricted in the distribution of their training datasets, and thus lack zero-shot generating ability Dhariwal & Nichol (2021). On the other hand, diffusion models Ho et al. (2020), especially text-to-image diffusion models Saharia et al. (2022); Rombach et al. (2022), are remarkable in synthesizing diverse and high-fidelity samples Ho et al. (2022) and composing unseen

images Liu et al. (2022). Hertz et al. (2022); Kwon et al. (2022); Wu et al. (2023a) have demonstrated that diffusion models possess a semantic latent space with word-to-semantic mappings, this property has been utilized for attention-map-based image editing and generation Cao et al. (2023); Chefer et al. (2023); Chen et al. (2024). Other related works Wu et al. (2023b); Wang et al. (2023) have considered the disentanglement in the latent space of the text embedding space, explored the feasibility of seeking disentangled latent space Lu et al. (2024); Everaert et al. (2023; 2024); Hahm et al. (2024), Gandikota et al. (2024); Sridhar & Vasconcelos (2024); Dalva et al. (2024) also considered the precise control of image semantics in the generative model, and proposed metrics for measuring counterfactual generation ability of generative models Monteiro et al. (2023); Zhou et al. (2023). However, the editing directions for precise and precise changing desired semantics are not explicit, posing a non-trivial challenge in manipulating semantics in the latent space.

### 6.3 Image Editing with Text-to-Image Diffusion Model

Recently, the capability of image editing of UNet-based Text-to-Image (T2I) diffusion models has been widely studied. Inspired by the semantic meaning of image-text attention maps, Hertz et al. (2022) demonstrates a word-to-semantic relationship in UNet-based T2I diffusion models, which motivates a series of image-editing methods based on manipulating attention maps Brooks et al. (2023); Cao et al. (2023); Chefer et al. (2023) or feature injection Tumanyan et al. (2023). Due to the entangled latent space of UNet-based diffusion model, these approaches often can only achieve coarse grained editing. Kawar et al. (2023); Wu et al. (2023a) proposes to optimize image embedding to achieve disentangled editing, Zhang et al. (2023); Zhao et al. (2025) utilize finetuning based method for fine-grained image editing. However, these approaches are not zero-shot and require extra fine-tuning on model parameters. To alleviate the entangled correlation between text tokens and image semantics, another type of methods Hertz et al. (2023); Nam et al. (2024) consider utilize score-distillation-sampling-based methods to iteratively update the noised image latents for precisely editing target items. While these works have improved the ability of T2I diffusion models to follow editing instructions, leveraging them for text-based precise image editing remains challenging. Our work verifies the disentanglement in the representation space of the diffusion transformer and thus enables precise image editing via latent space manipulation.

## 7 Conclusion

In this work, we explored the semantic disentanglement properties of the latent space of Text-to-Image diffusion transformers. Through extensive empirical analysis, we identified that the latent space of text-to-image DiTs is inherently disentangled, enabling precise and precise control over image semantics. Based on these insights, we proposed a novel Encode-Identify-Manipulate (EIM) framework that leverages this disentanglement property for zero-shot precise image editing. Our method demonstrates significant advancements over existing approaches, providing a robust and efficient solution for precise and precise image editing without additional training. Furthermore, we introduced a benchmark and a metric to quantitatively assess the disentanglement properties of generative models. We believe our work will inspire further research and serve as a valuable resource for advancing studies in generative models.

