# OpenReview forum: "Latent Space Disentanglement in Diffusion Transformers Enables Precise Zero-shot Semantic Editing"
_TMLR — Rejected by TMLR_

### Review · Reviewer_QJKA · 2024-12-14

**Summary Of Contributions:**

This paper introduces a simple yet effective framework, Encode-Identify-Manipulate (**EIM**), for achieving precise control over specific semantic attributes. EIM leverages the feature space of Diffusion Transformers to reveal their inherent semantic disentanglement properties. These properties enable independent control of various semantic attributes, facilitating precise image editing. Additionally, the paper proposes a novel metric, **SDE**, to measure latent space disentanglement, and introduces a new benchmark, **ZOPIE**, for comprehensive evaluation.

**Audience:**

Yes

**Broader Impact Concerns:**

No concerns.

**Claims And Evidence:**

Yes

**Requested Changes:**

- Include additional experimental results to demonstrate the generalizability of the method. For example, apply the same editing direction $n$ across different images and analyze the results.

- Provide a more thorough comparative analysis with related works. Suggested references:

  [1] Concept Sliders: LoRA Adaptors for Precise Control in Diffusion Models, ECCV 2024

  [2] Prompt Sliders for Fine-Grained Control, Editing and Erasing of Concepts in Diffusion Models

  [3] GANTASTIC: GAN-based Transfer of Interpretable Directions for Disentangled
  Image Editing in Text-to-Image Diffusion Models

- Add a comparison of computational costs with related methods to clarify the efficiency of the proposed approach.

- Provide additional details about the two prompts in Figure 5(a) to enhance clarity. Additionally, clarify whether the "degree: 0.8" in the prompts is necessary.

**Strengths And Weaknesses:**

##### Summary of strengths

- The paper presents a simple and effective method for image editing, addressing two key challenges: semantic loss and the insufficiency of manipulating only text embeddings. The method is novel and original.
- The paper is logically structured, and easy to follow.

##### Summary of weaknesses

- Some methods referenced in the figures are not adequately introduced, making it difficult for readers to locate the corresponding papers:
  - The proposed method in this paper is not explicitly named—does it refer to CSDS? Or should it be HSDS?
  - In Figure 12, the term "understate" is ambiguous—what method does it represent?
- The visualization results are not always satisfactory.
  - In Figure 6, under the +HSDS method, there are several inconsistencies:
    - The clothing style changes in the first row.
    - The second image in the second row shows a purple shirt.
    - In the fifth and sixth rows, there are style changes in the images.
  - These results are not sufficiently discussed in Section 4.2. Statements like "the image subspace should also be manipulated" are vague and lack clarity.

---

### Review · Reviewer_iSBj · 2024-12-16

**Summary Of Contributions:**

This paper investigates the disentanglement property of diffusion transformers and proposes the Encode-Identify-Manipulate (EIM) framework for zero-shot image editing. The EIM framework employs a Hessian Score Distillation Sampling (HSDS) method to identify the editing direction of the image based on the text and latent embeddings.

**Audience:**

Yes

**Claims And Evidence:**

No

**Requested Changes:**

See weaknesses

**Strengths And Weaknesses:**

Strengths:
1. The proposed investigation of semantic disentanglement in diffusion transformers is interesting and could provide insights for potential applications such as image editing.
2. The proposed method supports adjusting an intensity parameter to control the degree of editing, which is novel and practical in applications.

Weaknesses:
1. My biggest concern about this method is that the proposed framework does not actually enable precise zero-shot image editing as claimed in the paper. As shown in Figure 6, even after combining text embedding manipulation and HSDS, both foreground objects and background scenes change after editing (e.g. in the second last "robot" example, both the appearance of the robot body and the shelves in the background change completely). Recent zero-shot methods based on feature injection (e.g. PnP [1], Prompt-to-prompt [2]) or finetuning (e.g. MagicBrush [3], UltraEdit [4]) generally achieve better results in preserving the background details.
2. Can the author provide more details on how the baseline methods such as InstructPix2Pix are evaluated to achieve progressive transformations?
3. While the method cannot achieve precise image editing, given that the method enables editing images by manipulating image embeddings and is able to control the editing degree, one interesting exploration would be if the method can be used to refine the editing results produced by off-the-shelf image editing models, i.e. if the image editing results are too aggressive, if the method can generate interpolated results that preserve the desired edits but also reduces the artifacts. I am interested to see more results or potential applications related to manipulating the image embedding in this image editing framework.

Minor concerns:
1. Please consider simplifying the math notations in the manuscript as right now there are too many different but similar notations and it is hard to keep track of all of them.
2. The notations in the figures are not very clear. For example, in Figure 8, what are "CSDS", "Text", "Pix2Pix" and "Masa" referring to?

[1] Tumanyan, Narek, et al. "Plug-and-play diffusion features for text-driven image-to-image translation." Proceedings of the IEEE/CVF Conference on Computer Vision and Pattern Recognition. 2023.

[2] Hertz, Amir, et al. "Prompt-to-prompt image editing with cross attention control." arXiv preprint arXiv:2208.01626 (2022).

[3] Zhang, Kai, et al. "Magicbrush: A manually annotated dataset for instruction-guided image editing." Advances in Neural Information Processing Systems 36 (2024).

[4] Zhao, Haozhe, et al. "UltraEdit: Instruction-based Fine-Grained Image Editing at Scale." arXiv preprint arXiv:2407.05282 (2024).

---

### Review · Reviewer_cEnu · 2025-02-11

**Summary Of Contributions:**

This paper first shows that the semantic space of diffusion transformers is naturally disentangled compared to U-Net-based diffusion models.
Based on this insight, the authors propose to encode both the original prompt and the editing target prompt, subtract to find the difference, and then linearly interpolate along this idfference direction to effect the edits.
They apply this basic technique to both the text encoding and the image-based encoding across denoising time steps in transformer diffusion models.
They demonstrate how this works for more precise image editing while analyzing hyperparameters and other effects.

**Audience:**

Yes

**Broader Impact Concerns:**

It might be helpful to discuss uses of this that could be malicious, like faking someone's clothing or facial expression and using it to discredit or harm the person. Seems like a relatively minor broader impacts concerns but might be worth mentioning nonetheless.

**Claims And Evidence:**

Yes

**Requested Changes:**

- (**Most important requested change related to clarity of the claims/evidence**) A significant reorganization for clarity and flow is likely needed to improve the paper. Currently, the content and method seems interesting and insightful, but disjointed and difficult to follow logically.  Also, authors should work to emphasize the more important points and move less important points to the appendix.

     - See weaknesses above for more information on this point.
     - The introduction/contributions seems a little redundant where contributions are mentioned multiple times. Perhaps remove redundancy.
     - Figures should be explained before being shown in the paper. Many figures are only partially explained before being shown to the reader and this is confusing. Some figures are shown before there is any explanation. Sometimes the figures seem to be out of order. This makes it very difficult to read through the paper and results.

- Image editing metric could be compared to other metrics in the literature related to counterfactual image editing methods [Monteiro et al. 2023]. It seems that the goal is counterfactual-like image editing where only the specific attribute is changed while all others are maintained. Related metrics appear in a more theoretic paper [Zhou et al., 2024], where domain should change but class and other characteristics should not. It would be good to compare your metrics to these non-diffusion-based ones and show why your metrics are better suited.

Monteiro, M., Ribeiro, F. D. S., Pawlowski, N., Castro, D. C., & Glocker, B. Measuring axiomatic soundness of counterfactual image models. In The Eleventh International Conference on Learning Representations. 2023.

Zhou, Z., Bai, R., Kulinski, S., Kocaoglu, M., & Inouye, D. I. Towards Characterizing Domain Counterfactuals for Invertible Latent Causal Models. In The Twelfth International Conference on Learning Representations. 2024.

- What is proposition 1 in Section 2.2 (referenced in Section 5.2)? I do not immediately see it. If you have a formal proposition, you need to set it apart from the rest.

Minor comments:

- It seems that you modify the direction of the image in a special way using HSDS while for text you simply take the difference since it is fixed throughout the denoising process. Is that correct? Essentially, for text, it is simple. For denoising, you have to determine the right direction for each timestep and want some sort of consistency among time steps.

- Figure 2b looks like it just copies the reference image rather than maintaining the semantics of the original image. Seems like reference-based image editing is less capable.

- The justification for natural disentanglement (Fig 4) seemed a bit vague. At a high-level, I can see that cross attention vs self-attention could yield different things but the intuition is not explained well and there is only a small description of this phenomena in the main text.

**Strengths And Weaknesses:**

**Strengths**
- Interesting research question about the joint latent space of text and image tokens.

- Useful insight into the disentanglement of the space.

- Simple and straightforward zero-shot approach to diffusion-based image editing with transformer-based modelsj.

- Nice looking qualitative results that showcase the proposed method.

- Results are relatively extensive exploring different aspects of the method including hyperparameter analysis.

**Weaknesses**
- The observations and organization of the paper is challenging to follow (particularly Section 2) and jumps around a fair bit. While Section 2 showing some interesting properties of diffusion transformers and demonstrating some failure cases is relatively interesting and insightful, it does not flow well and figures are referenced in different orders and interleaved together. This section could use a significant reorganization for better flow and clarity.

- Similarly, the experiments could be better integrated. Rather than explaining the whole setup and then making observations. I think it would be more effective to organize this section according to the points you want to make. Perhaps research questions or claims. Only describe the necessary setup as needed inline.  Perhaps something like: Research question/claim, explanation of how you will measure this, results presentation and then discussion of results. But do this grouped by claim/hypothesis rather than by "Qualitative" vs "quantitative". There are many experiments so it's challenging to keep track.

- Along those lines, perhaps some of the results could be put in the appendix to avoid a long list of items. It would be better to focus on the most important ones, briefly mention the others and then put the full results in the appendix for the interested reader. As it stands, it is difficult to understand what is the most important and impactful experiments and they all seem to blend together---ultimately hindering the impact of the results section.

- The identify stage seems relatively similar to the ideas suggested in Section 2. This is fine but confusing why you add it again here. Again, organization could help avoid this redundancy and confusion.

- The explanation of HSDS is confusing to read. Perhaps you need some background on SDS. It is important to identify what the key difference between SDS sampling and your HSDS sampling is. The motivation " this might incorrectly affect the semantics controlled by text embeddings" is imprecise. This is too vague. Perhaps showing a direct comparison of SDS update vs your HSDS update would make the distinction more clear. Also, what do the two terms in 1 mean?  How does this connect to the direction $n$ mentioned before? It seems like this is some time-based version of your original idea of going in the $n$ direction but across multiple timesteps now.

- A clear difference from prior works would help the paper. The U-Net vs transformer part is clear but that seems to already be known. Can you highlight what was not known or noticed in prior works? More generally, can you pinpoint your difference from prior work in terms of methodology? The "diff" from prior works was not clearly explained overall.

---

### Public Comment · ~Shiwen_Zhnag1 · 2025-02-12
**Very surprised about not discussing a closely related work**

The CVPR 2023 paper "Uncovering the Disentanglement Capability in Text-to-Image Diffusion Models" [1]  proposes very similar findings and embedding manipulation methods compared to this paper. In fact, the method is almost is same as that in this paper:
1. [1] discovered the disentanglement property of stable diffusion by replacing the textual embedding, as with detailed discussions in 3.2. I do not find any disentanglement materials in this paper significantly differed from section 3.2 in [1]
2. [1] proposes linear interpolation method for control strength of editing in Eq.5. which is given by $\mathbf{c}_t = \lambda_t \mathbf{c}^{(1)} + (1 - \lambda_t) \mathbf{c}^{(0)} \equiv \mathbf{c}_t^{(\lambda)}, \quad$, and this is **exactly the same** as the core interpolation method in this paper.
3. [1] applies this method for editing as in section 3.4, and this paper also only considers image editing, but not **compares with** [1]
**Surprisingly**, this paper cites [1], and briefly talked about it in the related work. I am looking forward to hearing back from the authors

[1] Wu, Qiucheng, et al. "Uncovering the disentanglement capability in text-to-image diffusion models."  CVPR 2023

---

> ### Public Comment · ~Shiwen_Zhnag1 · 2025-02-14
> **More things to notice**
>
> What makes the entire contribution, validity and correctness of this paper more dubious is Table 4. In Table 4., the authors actually mentioned [1]. However the description about [1] in the table is ***completely wrong***. The authors claimed that [1] is not disentangled, without linearity (the linear combination of features), not zero-shot, but every of these claims are wrong.
>
> I acknowledge [1] does not use the HSDS method for score editing. But HSDS closely resembles the DDS/SDS method in [2]. The only possible methodological contribution of this paper should be new H part of HSDS. However, authors do not provide any ablation studies on adding the Hessian part. In addition, the authors do not compare the DDS/SDS add-on with their proposed (H)SDS add-on. It is very challenging to figure out whether their proposed (H)SDS actually leads to any significant improvement given that their visual results still show some significant background semantic change, as pointed out by some reviewers.
>
>
> [1] Wu, Qiucheng, et al. "Uncovering the disentanglement capability in text-to-image diffusion models." Proceedings of the IEEE/CVF conference on computer vision and pattern recognition. 2023.
>
> [2] Hertz, Amir, Kfir Aberman, and Daniel Cohen-Or. "Delta denoising score." Proceedings of the IEEE/CVF International Conference on Computer Vision. 2023.

---

> ### Author Response · Authors · 2025-02-26
>
> Dear Shiwen,
>
> Thank you again for your interest in our group’s work. However, some of your points seem to stem from misunderstandings.
>
> For point 1: Our paper specifically investigates the disentanglement properties of diffusion transformers, while [1] focuses primarily on UNet-based models—a distinction we clearly emphasize in the introduction. Transformer-based models employ fundamentally different mechanisms for aligning textual conditions with image embeddings, resulting in different disentanglement properties compared to those observed in UNet-based diffusion models. Moreover, our paper offers a comprehensive discussion on the origin, application, and empirical findings related to the disentanglement property in diffusion transformers (see Section 2), whereas [1] only demonstrates a weak disentanglement effect without thorough analysis.
>
> For point 2: Leveraging the disentanglement property of diffusion transformers, we introduce a zero-shot image editing method capable of precisely modifying semantic intensity. In contrast, [1] does not target this type of editing. Additionally, note that (1) the $\lambda_t$ in [1] is a learnable parameter generated by a network module, and (2) it is not intended for adjusting semantic intensity. Furthermore, in [1] the motivation behind introducing $\lambda_t$ is to determine the optimal point for replacing $c_0$ with $c_1$ in a soft way.
>
> For point 3: Our method is distinctly zero-shot, enabling precise modification of semantic intensity. Conversely, [1] relies on a trainable module, and thus should be a non-zero-shot approach. These methods operate under entirely different settings.
>
> For point 4: As stated in Section 4 of [1], their method is not zero-shot. Regarding linear manipulation and disentanglement, there might lack sufficient evidence to show that their method can precisely edit semantic intensity using only a linear combination of features in a zero-shot manner.
>
> For point 5: The primary objective of our paper is to uncover and analyze the semantic disentanglement property in transformer-based diffusion models—not merely to propose an image editing method. This broader perspective distinguishes our work from previous approaches and underscores its significance. Our image editing task is a direct application of the discovered disentanglement property, allowing us to modify the intensity of a specific semantic attribute (for example, adjusting the degree of a person’s smile) while preserving most irrelevant features. We acknowledge that some unintended variations in image details may occur, however, our method is designed to focus on controlled modifications of the target attribute rather than enforcing strict invariance across all regions.
>
> Best regards,
>
> Authors of Paper Submission 3640

---

> ### Public Comment · ~Shiwen_Zhnag1 · 2025-02-26
> **Factual Errors**
>
> Thanks authors for responding to my concerns. There are some factual errors in the authors' response:
>
> 1. [1] is indeed a zero-shot method. The opimization ("training" as you mentioned in the response") in [1] is only for finding a $\lambda$ balancing between preserving original content and conforming to the target style. This is clearly stated in section 3.3: ```"Our goal is to find an optimal 1:T such that $X_0^{\lambda}$ maintains the same semantic content as $X_0^{0}$, but conforms to the style described in c, which is achieved by solving the following optimization problem"``` In addition, [1] does not require any training data. The "training" in [1] means that if they find a good $\lambda_t$ for one image, then can that optimized $\lambda_t$ be transferred to another image to reduce computational cost. It does not mean that [1] is not zero-shot.
>
> 2. $\lambda_t$ is not an output "generated by a network module". It is just a scalar for optimization (an input to the optimization problem). This is stated in Appendix Table.2
>
> 3. "It is not intended for adjusting semantic intensity. ", This is also wrong. In [1] section 3.3, they clearly state that adjusting $\lambda_t$ largely causes more semantic change and losing fidelity to original image.
>
> Some other problems:
>
> 1. Table 4 in this paper is still not changed in the revision.
>
> 2. If your method has a better disentanglement than [1], why don't you compare with [1] qualitatively or quantitatively?

---

### Decision · Action_Editor_nqhg · 2025-03-22

**Recommendation:** Reject

**Comment:**

The decision to recommend rejection stems from critical unresolved issues that undermine the paper’s validity and impact.

1. The core claims lack sufficient differentiation from prior work, particularly the CVPR 2023 paper [1], which proposes nearly identical methods for disentanglement and embedding interpolation, yet the authors fail to rigorously address this overlap beyond superficial citations.

2. The paper’s organization is disjointed, with redundant sections (e.g., Section 2 vs. the “identify stage”), unclear technical explanations (e.g., HSDS vs. SDS), and poorly structured experiments that obscure key contributions. Methodological weaknesses—such as unexplained parameters (“degree: 0.8”), missing computational cost analysis, and unconvincing claims of “precise editing” (e.g., background artifacts in Figure 6)—further weaken the technical rigor.

While the topic of diffusion transformer properties and embedding-based editing holds potential interest for TMLR’s audience, the authors’ minimal rebuttal effort and failure to clarify reorganization plans or novelty leave reviewers unconvinced. To salvage the submission, reviewers emphasize the need to rigorously compare with [1] and U-Net-based methods, restructure the paper for clarity, justify methodological choices, expand evaluations (e.g., background preservation, applications in refining existing editors), and provide concrete evidence to support claims. Addressing these issues could transform the work into a meaningful contribution, but in its current state, it falls short of acceptance criteria.

**Audience:**

Some individuals in TMLR’s audience would find the findings interesting. The paper’s exploration of diffusion transformers’ disentanglement properties and embedding-based editing aligns with active research areas in text-to-image generation. Specific points of interest include:

1. The potential to refine existing editing methods (e.g., interpolating aggressive edits from tools like InstructPix2Pix).

2. Insights into diffusion transformer behaviors compared to U-Net architectures.

However, the current presentation and lack of novelty relative to prior works (e.g., [1]) limit broader appeal. Addressing comparative analysis and methodological clarity would strengthen relevance.

**Claims And Evidence:**

The claims in the submission are not fully supported by accurate evidence. Key issues include:

1. Insufficient comparative analysis: The authors claim that they propose a new method conduct a comprehensive evaluation. However, they fail to rigorously compare its method with critical prior works (e.g., CVPR 2023 [1], Concept Sliders, GANTASTIC) that propose nearly identical techniques for disentanglement and embedding manipulation. Also, the authors’ response was deemed minimal and failed to clarify reorganization plans or address key concerns like methodology overlap with [1].

2. Lack of methodological clarity: The necessity of parameters (e.g., "degree: 0.8") and the distinction between HSDS and SDS are poorly explained. Reviewers noted vague motivations and insufficient technical comparisons.

3. Weak empirical validation: The evaluation is limited (e.g., no computational cost analysis, superficial discussion of visualization inconsistencies) and does not adequately address precision in zero-shot editing. For example, Figure 6 shows unintended background changes, contradicting claims of "precise editing."